# The State-of-the-Art of the Humoral Memory Response to Snakebites: Insights from the Yanomami Population

**DOI:** 10.3390/toxins15110638

**Published:** 2023-11-01

**Authors:** Sewbert Rodrigues Jati, Thais Andréa dos Anjos Martins, Anderson Maciel Rocha, Guilherme Melo-dos-Santos, Isadora Sousa de Oliveira, Isabela Gobbo Ferreira, Altair Seabra de Farias, Eloise T. M. Filardi, Felipe Augusto Cerni, Marco Aurélio Sartim, Jacqueline de Almeida Gonçalves Sachett, Wuelton Marcelo Monteiro, Manuela Berto Pucca

**Affiliations:** 1Graduate Program in Tropical Medicine (PPGMT), State University of Amazonas, Manaus 69850-000, Brazil; sewbert@gmail.com (S.R.J.); thaisamartins96@hotmail.com (T.A.d.A.M.); amr.0609@gmail.com (A.M.R.); asfarias@uea.edu.br (A.S.d.F.); felipe_cerni@hotmail.com (F.A.C.); marcosartim@hotmail.com (M.A.S.); jac.sachett@gmail.com (J.d.A.G.S.); wueltonmm@gmail.com (W.M.M.); 2Department of Teaching and Research, Dr. Heitor Vieira Dourado Tropical Medicine Foundation, Manaus 69850-000, Brazil; 3Department of Education and Sports of Roraima, Boa Vista 69301-130, Brazil; 4Graduate Program in Bioscience and Biotechnology Applied to Pharmacy, School of Pharmaceutical Sciences, São Paulo State University (UNESP), Campus Araraquara, São Paulo 19060-900, Brazil; gmelo5921@gmail.com (G.M.-d.-S.); e.filardi@unesp.br (E.T.M.F.); 5Department of BioMolecular Sciences, School of Pharmaceutical Sciences of Ribeirão Preto, University of São Paulo, Ribeirão Preto 19040-903, Brazil; isadora_so@yahoo.com (I.S.d.O.); igobboferreira@yahoo.com.br (I.G.F.); 6Department of Biotechnology and Biomedicine, Technical University of Denmark, 2800 Kongens Lyngby, Denmark; 7Pro-Rectory of Research and Graduate Studies, Nilton Lins University, Manaus 69850-000, Brazil; 8Department of Clinical Analysis, School of Pharmaceutical Sciences, São Paulo State University, Araraquara 19060-900, Brazil

**Keywords:** snakebite immunity, Yanomami, antivenom, snakebite resistance, immune response, Amazon

## Abstract

Snakebite envenomation (SBE)-induced immunity refers to individuals who have been previously bitten by a snake and developed a protective immune response against subsequent envenomations. The notion stems from observations of individuals, including in the indigenous population, who present only mild signs and symptoms after surviving multiple SBEs. Indeed, these observations have engendered scientific interest and prompted inquiries into the potential development of a protective immunity from exposure to snake toxins. This review explores the evidence of a protective immune response developing following SBE. Studies suggest that natural exposure to snake toxins can trigger protection from the severity of SBEs, mediated by specific antibodies. However, the evaluation of the immune memory response in SBE patients remains challenging. Further research is needed to elucidate the immune response dynamics and identify potential targets for therapeutic interventions. Furthermore, the estimation of the effect of previous exposures on SBE epidemiology in hyperendemic areas, such as in the indigenous villages of the Amazon region (e.g., the Yanomami population) is a matter of debate.

## 1. Introduction

In Brazil, approximately 30,000 snakebites are reported each year [1], presenting high morbidity and mortality rates within the Brazilian Amazon region [2]. Snake venoms are composed of a protein cocktail of high complexity and diversity [3], triggering a variety of biochemical and toxicological effects on victims, which influences several clinical manifestations [4] ranging from mild to severe outcomes, including hospitalization for long periods, surgical procedures, and follow-up for rehabilitation [5]. Indeed, snakebite after-effects depend on several factors, including the species, region of the bite, quantity of venom injected, and influence of ecology and evolution in driving inter- and intra-specific venom variations, in addition to the health condition of the victim [6,7].

Antivenom (i.e., horse-derived polyvalent antibodies) has been the primary and single specific snakebite treatment for more than a century. Although lifesaving, antivenoms still have therapeutic limitations [8,9], presenting limited efficacy against some effects of envenoming, such as local tissue damage [4,10].

Based on that, vaccines targeting snakebites were raised as a possibility to induce protection to those at risk for death by snakebite [11]. Knowing that many snake venom-derived toxins are immunogenic, different experimental approaches and studies have demonstrated that vaccine-elicited antibodies can neutralize venoms and protect from envenomation injury [12,13,14,15]. In humans, the ability of venom to induce neutralizing antibodies has also been demonstrated [16]. This article will review evidence of human antibodies targeting snake venoms and inducing protection for the victims, aiming to understand the memory immune response.

## 2. Snakebites in the Yanomami Indigenous Community of the Brazilian Amazon

The Yanomamis are a hunter–horticulturist indigenous population from the interfluvial tropical forest of the western Guiana massif, who inhabit the borders between Venezuela (upper Orinoco and Cassiquiare) and Brazil (upper *Rio Branco*, left bank of *Rio Negro*). They constitute a cultural and linguistic set composed of four territorially adjacent subgroups that speak mutually intelligible languages: the Yanomami (approximately 56% of the ethnic group), the Yanomam (25%), the Sanumá (14%), and the Ninam (5%) [17]. 

On the Brazil side, the Yanomami Indigenous Land (YIL) was demarcated in 1992, occupying 96,560 km^2^ in the west of the state of Roraima and north of the state of Amazonas (Figure 1), where around 21,600 indigenous people live in 260 communities [18].

Hence, the Yanomami people’s way of life, deeply intertwined with the *Urihi* (Forest-Earth, as it is referred to in Yanomami language), renders them particularly susceptible to snake-related accidents. Their homes (called *Xapono* in the Yanomami language) lack walls that would provide insulation from the surrounding forest. Additionally, their swidden agriculture is carried out in dense forest, and they frequently visit collection sites to gather roots and other plant materials essential to their daily lives. Moreover, their hunting and fishing grounds, crucial for sustenance, expose them to further potential encounters with snakes. 

Since the Yanomamis are a closed indigenous community, there is no road to access the YIL, resulting in communities with difficult access (remote), which are marked by frequent snakebite envenomings due to the typical humid tropical Amazon Forest environment. Although highly prevalent, snakebites in YIL are commonly treated in situ, with the victim’s care mainly performed by local healers and health professionals working in the area [19]. In cases of severe envenomings with no improvement following local therapy, the indigenous victims can be transferred by air to main hospitals in cities [20,21]. Although snakebites are considered one of the main diseases in the YIL, there is a lack of studies in the region compared with other regions in the Amazon such as the state of Pará [22,23,24,25] and the Central Amazon [26,27].

As the Yanomamis live in places with difficult access, different causes have been shown to be responsible for the negligence of snakebite care, including the delay in therapy and difficulty in removing patients from the isolated areas, resulting in a higher lethality rate and sequelae than in other parts of Brazil [22]. 

In addition, recent studies [18,28,29] corroborate the fact that the Yanomamis have a mythological relationship with snakes, which can be considered as an “evil” for their population. Thus, they believe that being attacked by a snake may be a kind of punishment by the gods for some act performed. As a result, many Yanomamis do not seek medical assistance following snakebites, which, consequently, may result in the worsening of the envenomation. On the other hand, many Yanomamis that do not seek any care following a snakebite based on punishment beliefs testify that they had recovered from the envenoming without any kind of treatment, raising the possibility of a memory immune response. Nevertheless, the memory response targeting snakebites in the Yanomamis has not yet been studied or demonstrated. Nevertheless, it is undeniable that coexistence between the Yanomamis and snakes has persisted for several generations of this ethnic group.

Detailed clinical-epidemiological data obtained from the Yanomami and Ye’kuana Indigenous Special Health District (DSEI-YY) from 2017 to 2022, conducted in accordance with the Declaration of Helsinki (protocol approved by the Research Ethic Committee of the Federal University of Roraima under number CAAE 53970721.4.0000.5016), performed by the research group and presented for the first time here, demonstrated that:(i)In the YIL, men are mainly affected by snakebites at an average of 61.3%, which is statistically different in comparison to women (paired *t*-test, *p* = 0.034). However, compared with other studies [30,31], the percentage of Yanomami women (38.7%) who were affected by snakebites is 4.85 times higher than the national average (8%), probably due to the women-related activities in the community, as they also work in the field and harvest firewood, food, and medicinal products in the forest.(ii)Most snakebites occur in the age group of 20 to 39 years old (35.2%), followed by 15 to 19 (17.6%), 10 to 14 (16.9%), 40 to 59 (16.0%), 5 to 9 (10.4%), 60 to 79 years (2.1%), up to 4 years old (1.3% of cases), and over 80 years old (0.5% of cases), similarly to the results obtained by the pioneering Vital Brazil studies [30] and others [31,32,33].(iii)Regarding seasonality, the month with the highest snakebite incidence was May (mean of 19.8) and the one with the lowest incidence was December (mean of 9.5) (Figure 2). This is a different result compared to a previous study that marked July as the month with the highest occurrence of snakebites and October as the one with the lowest occurrence.(iv)The highest number of snakebites were reported in the municipality of Alto Alegre in the Roraima States/Brazil (50.5%), which was expected since the region includes the Serra dos Surucucus, located in the westernmost portion of this municipality, in the border region with Venezuela, known to be inhabited by snake-rich fauna [31].

Snakebites can be life-threatening and reach a lethality of 2.75% in the YIL, which is five times higher than the lethality rate in Brazil (0.6%) [33]. Concerning the snakebite prevalence per 100,000 people, the prevalence of snakebites in the YIL is 69.7/100,000, which is much higher than the Brazilian prevalence of 1.2/100,000 inhabitants. Thus, we can determine the relative risk for the Yanomami compared to the Brazilian population as follows:

Brazilian lethality rate: 0.6% (0.006)

Brazilian snakebite prevalence: 1.2 per 100,000 (0.0012)
Risk of death for Brazilians=0.006×0.0012

Yanomami lethality rate: 2.75% (0.0275)

Yanomami snakebite prevalence: 69.7 per 100,000 (0.0697)
Risk of death=0.0275×0.0697
$$Relative\;risk=\frac{Risk\;of\;death\;for\;Yanomami}{Risk\;of\;death\;for\;Brazilians}$$
$$Relative\;risk=\frac{0.0275\times 0.0697}{0.006\times 0.0012}=12.7$$

This means that the Yanomami population has a risk of death from snakebites that is approximately 12.7 times higher than the risk for the Brazilian population. Despite the Yanomami having a higher lethality rate, due to the higher incidence of snakebites (58 times higher), we expected that proportionally the lethality should be even higher, especially due to the lack of health assistance, but the lower death rate than expected lead to the question of whether they may have developed some form of mechanism to control snake venom in their bodies. Notwithstanding, the severity and long-last disabilities in the Yanomami population have also been documented [21], indicating that snakebites have a significantly greater impact on the Yanomami population compared to other groups. 

The Yanomami people are particularly noteworthy as they have had recent contact with external societies. The *Fundação Nacional dos Povos Indígenas* (FUNAI) defines “recent contact” indigenous groups as those who maintain both permanent and/or temporary connections with segments of the national society. Regardless of the duration of contact, these groups exhibit distinct characteristics in their relationship with the national society, displaying selectivity and autonomy in their incorporation of goods and services. As a result, they preserve their unique social structures and collective dynamics, and exercise a high degree of autonomy in defining their interactions with the state and national society [34].

## 3. The Immunological Response Targeting Snake Venoms

After a snakebite and the subsequent introduction of venom, the immune system is initiated in accordance with a well-established pattern, leading to the activation of both innate and adaptive immune responses. It is worth emphasizing that this immune mechanism is applicable to any other external or extracellular antigen (Figure 3) [35]. Briefly, antigen-presenting cells (APCs) are recruited to the bite site, where they recognize and process venom-derived toxins into small peptides. These peptides are then presented on the cell membrane by the major histocompatibility complex (MHC). Stimulated by the cytokines produced during the establishment of the innate immune response, the APCs migrate to the regional lymph nodes, where naive T CD8+ and T CD4+ cells recognize the antigen-MHC complex. Upon lymphocyte activation, they differentiate into effector lymphocytes. Simultaneously, naive B cells located in the lymph nodes recognize the toxin-derived components of the venom, known as antigenic epitopes. The B cells present it again on the cell membrane bound to MHC class II molecules. The subsequent interaction between the antigen-MHC class II complex on the B cell membrane and the activated helper T cell generates a second activating signal within the B cell. This signal triggers their proliferation and subsequent differentiation into plasma cells and memory cells leading to the production of specific antibodies targeting venom-derived toxins [36,37]. Memory B cells are predominantly localized in secondary lymphoid organs, such as lymph nodes and spleen. The role of memory B cells is to provide a rapid and enhanced immune response upon re-exposure to a previously encountered antigen. They are responsible for the long-term maintenance of immunological memory. Short-lived plasma cells are predominantly localized in the medullary cords of lymph nodes and the red pulp of the spleen and play a crucial role in the early immune response by producing high levels of IgM and IgG antibodies, which are important for the initial control of infections. On the other hand, long-lived plasma cells primarily reside in the bone marrow. They are responsible for maintaining a sustained production of IgG antibodies over a prolonged period [37].

To date, the only strategy available for assessing the memory response to snakebites is by measuring the humoral response through the quantification of venom-specific antibodies in serum. This approach involves determining the levels of venom-specific IgM and IgG antibodies using techniques such as enzyme-linked immunosorbent assay (ELISA). By measuring the antibody titers in the serum, valuable information can be obtained regarding the presence and magnitude of the immune memory generated after a snakebite. However, assessing other aspects of the memory response, such as the presence and functionality of memory cells (e.g., effector memory cells) and the dynamics of long-lived and short-lived plasma cells, poses challenges due to technical limitations and the complexity of accessing these cell populations in living human subjects [38].

## 4. Evidence of Humoral Memory Response following Snakebite Envenomings

Despite in vivo evidence demonstrating the capacity of humoral adaptive immunity to neutralize myotoxins and other compounds responsible for local and systemic venom symptoms [39,40,41], the delayed timeframe between venom inoculation and the production of sufficient quantities of antibodies for toxin neutralization poses a limitation for an immediate beneficial effect of the humoral response following snakebite envenomation. In fact, venom effects initiate immediately after the snakebite and are subsequently intensified by the body’s inflammatory response, resulting in significant symptoms within a few hours and potentially leading to fatality. On the other hand, the entire duration, from the snakebite incident to the immune activation, and the final production of specific antibodies against the toxins, takes a minimum of 5 to 7 days (Figure 4) [37].

In 1977, Theakston, Lloyd-Jones, and Reid, from the Liverpool School of Tropical Medicine, identified high levels of antibodies against the venoms of *Echis carinatus*, *Bitis arietans*, *Causus maculatusan* in rabbits, using as their method an enzyme-linked immunosorbent assay (ELISA) test [42]. After the initial study, the English research group started to analyze the immune response in two different populations: Waorani indigenous people from Ecuador and Nigerians from Bambur in the Benue Valley. First, the researchers analyzed the immune response of 223 indigenous people and revealed a high incidence (79% of the sample) of victims exhibiting antibodies against snake venom, with most individuals displaying reactivity against multiple species. In the second study, 347 subjects with alluding exposure to snakebite were tested for antibodies against the most important local species, with 37% displaying venom antibodies [16].

Subsequently, a follow-up study was conducted using sera with antibodies identified from seven indigenous Waorani people. In samples with high antibody concentrations, the researchers explored the potential for those antibodies to neutralize the effects of a second exposure, using animals. They observed that rodent groups exposed to both immunoglobulin-rich serum and a lethal dose of venom exhibited a higher survival rate compared to those exposed to sera with lower antibody concentrations or the control group [43]. These findings suggested that humoral memory could confer protective benefits in cases of subsequent snakebite accidents caused by the same species/genus, although the existing laboratory evidence in humans remains limited.

Subsequent studies have investigated the dynamic development of the adaptive humoral immune response following envenomation. These investigations have involved the meticulous measurement of isolated or prospective serum samples obtained from individuals who have experienced snakebites (Table 1). Consistently, these studies have revealed the production of diverse classes of immunoglobulins in response to venom exposure. Among these immunoglobulins, the initial one identified is IgM, which can be detected using the ELISA method as early as the third day post-exposure. However, IgM production gradually declines over the subsequent weeks. The kinetics of IgM production are relatively short-lived, resulting in lower levels compared to IgG, the latter being detectable and capable of persisting for several decades in certain individuals following venom exposure. Remarkably, individuals with a history of previous envenomation display earlier detectable IgG levels, characterized by higher concentrations and a more pronounced peak of production when compared to patients without prior exposure [44,45,46].

Patients who have not been bitten by snakes can also develop an immune response against snake venom. For instance, in 1987, Wadeen and Rabson demonstrated the production of high titers of IgE and IgG antibodies, following exposure (whether through dermatological or inhalation contact) to *Hemachatus haemachatus*, in a snake farm worker [50]. This finding indicates an induced immune response that can occur after any type of exposure [55,56]. 

Furthermore, it has been observed that the production and maintenance of the venom-specific antibody pool can be influenced by various factors. For instance, the administered venom dose has been shown to impact the magnitude and kinetics of the humoral immune response [45,46]. Additionally, the administration of antivenom and the presence of uremia have been found to affect the production and sustainability of the antibody response [46]. However, it is important to note that conflicting evidence exists regarding the influence of the volume of antivenom administered on the evolution of the humoral response against venom. Further research is needed to elucidate the complex interplay between these factors and their specific impact on the dynamics and effectiveness of the humoral immune response in the context of snake envenomation. Such knowledge would contribute to improving the understanding of the immune mechanisms involved, and aid in the development of more effective therapeutic strategies for snakebite management.

## 5. Vaccine Approaches for Populations Living in Areas of Snakebite Risk

The optimal preventive measure against snakebites is to avoid encounters with snakes. Nevertheless, accidental encounters still occur in most situations, leading to envenomation and necessitating the best approach for overcoming it, which is treatment with antivenom. In 1877, Sewall conducted the first investigation on a prophylaxis for snakebite envenomation, subcutaneously inoculating rattlesnake venom into pigeons repeatedly. The observed signs and symptoms in the animals ranged from paw paralysis to death, occurring 3 to 20 h after injection. However, the study demonstrated that when animals received repeated venom injections they evolved to a “resistance” to the venom’s harmful effects, recovering quickly from the venom’s effects, even when exposed to lethal or higher doses, although the resistance lasted no longer than five months, as observed in one of the tested animals [57]. 

Later, Wallis and Wallis (2005) described the vaccination of dogs against rattlesnake venoms, showing that the dogs’ produced antibodies were able to bind and neutralize the venom effects of *Crotalus atrox* in vitro and in vivo [12]. Based on this information, a vaccine specifically designed for *C. atrox* has been developed and is currently available through Red Rock Biologics (http://redrockbiologics.com, accessed on 28 August 2023) for both dogs and horses. However, a study conducted by Leonard, Bresee, and Cruikshank (2014) suggests that there was no discernible difference in terms of morbidity and mortality rates between dogs that were vaccinated and those that were not [58]. Another study reports the vaccination of cattle with *Bothrops asper* venom, showing that the systemic effects that occur during moderate envenoming were prevented, and coagulopathies were delayed, in relation to non-vaccinated animals; however, the infusion of antivenom was still required [59].

Active immunization was tested not only with venoms, but also with isolated toxins derived from venoms. A study using α-cobratoxin from *Naja kaouthia* observed that the animals that were immunized with the toxin and later challenged with it survived, avoiding lethality [11]. Similarly, mice immunization with α-bungarotoxin peptides from *Bungarus multicinctus* led to a protective effect, even using high doses of the toxin [60].

These studies are not restricted to snake venoms, but also are conducted on toxins from other animal venoms. Costa et al. (2020) produced monoclonal antibodies able to recognize metalloprotease of *Loxosceles* spp. spider venom, through the immunization of mice with a recombinant multiepitopic protein derived from loxoscelic toxin (rMEPLox), and observed that the monoclonal antibodies produced were also able to neutralize the fibrinogenolytic activity of *L. intermedia* spider venom [61]. Moreover, Cerni et al. (2023) reported that the immunization of mice with Ts5, a toxin from *Tityus serrulatus*, was able to trigger the production of antibodies capable of inhibiting the pain caused during the scorpion envenomation [62]. 

In the context of human immunization against snake venoms, significant observations have been made. Haast and Winer (1955) conducted a study on a man who deliberately self-immunized using Elapidae venoms and managed to survive a snakebite incident. The individual described his symptoms from the time of the bite until his arrival at the hospital. Surprisingly, by the third day following the accident, he had fully recovered and was able to resume work the next day [63]. Another case study presented by Watt, Parrish, and Pollard (1956) involved a herpetologist who, over a period of 12 years, was bitten by ten venomous snakes and, remarkably, survived each encounter [64]. Building upon this line of research, Parrish and Pollard (1959) examined North American pit viper bites in 14 patients who had experienced multiple bites. Contrary to expectations, the study found that these repeated bites did not induce permanent immunity. In fact, some cases resulted in hypersensitivity reactions, which could potentially lead to fatal outcomes [65].

Flowers (1963) documented the immunization process of an individual who received 17 injections of *Naja naja* venom over a period of five months. After the immunization period, the individual’s antibodies were used in neutralization assays in mice, where 1 mL of his serum was able to neutralize 0.244 mg of venom [66]. Interestingly, despite having received 17 doses of venom, the individual continued to receive monthly injections of 5 mg of venom. However, during the study, approximately two weeks after the 24th venom injection, an accidental snakebite occurred. The snake responsible for the bite was identified as *N. naja*. Despite being admitted to the hospital 15 min after the bite, the individual did not receive antivenom therapy as he had already received venom doses for his immunization. Notably, he presented only mild symptoms of envenomation, and experienced only pain, swelling, and necrosis in the affected finger [67]. These human studies, involving repeated venom injections, have provided valuable insights that allow us to conclude that individuals can develop immunity to venom, rendering them protected against future bites [68].

However, it could be argued that the largest and most noteworthy study was the exceptional immunization campaign of the Amami and Okinawa Islands (Japan) performed over a period of three years (1957–1960), involving over 40,000 volunteers, aiming to immunize against *Trimeresurus flavoviridis* venom. This pioneering study evaluated the effectiveness of the immunization protocol in preventing local lesions caused by the venom. Meticulous data collection and analysis revealed that the immunization successfully prevented the occurrence of local lesions in vaccinated individuals exposed to the venom, highlighting the success of large-scale efforts in addressing the devastating impacts of snakebite envenomation and providing valuable insights for future research and the development of effective immunization strategies [69].

Despite the partial effectiveness observed in these human studies, it is important to highlight the cost-effectiveness of vaccines. In Brazil, snakebite care costs USD 78.15 per patient daily, including the price of the antivenom and the indirect care [70]. On the other hand, in sub-Saharan Africa the average cost of an antivenom vial is approximately USD 124 [71], which, although expensive, is relatively more affordable compared to vaccine production in low-income countries [72]. Thus, while previous studies have explored vaccination strategies, it is essential to acknowledge that these approaches have many limitations for effectively mitigating the risks associated with snakebites. The incidence of snakebites is comparatively lower than that of other epidemic diseases. Moreover, developing snakebite-specific vaccination strategies for various venomous snake species can be cost-prohibitive and may lack cost-effectiveness. Furthermore, snakebites can result from numerous snake species, each with distinct venom compositions, which should reveal the financial constraints associated with the development of multiple snake-specific vaccines and the challenges in providing widespread coverage for a relatively rare event like a snakebite. 

## 6. Snakebite Immunity: A Fact Lacking Precise Assessment

According to the scientific evidence discussed in this study, the concept of snakebite-induced immunity is clear but very complex. While some studies suggest that individuals with a history of snakebites may exhibit partial or complete protection, or reduced severity of symptoms upon subsequent envenomation, the extent and duration of this immunity remain unclear.

Factors such as the snake species, venom composition, individual immune response, and time elapsed since the previous bite can greatly influence the outcomes. Indeed, it is worth noting that while snakes may share similar classes of toxins, we cannot assume that immunity to the venom of one species extends to another, even within the same genus and species, due to the venom variability [73]. Moreover, a previous encounter with a venomous snake does not necessarily indicate the activation of humoral immunity, particularly when there are no symptoms of envenomation, suggesting the possibility of a dry bite [74].

One of the challenges in assessing snakebite-induced immunity is the lack of standardized protocols and controlled studies. Many of the reported cases are anecdotal, making it difficult to draw definitive conclusions. Furthermore, snake venoms are highly diverse, with variations in their composition even within the same species. This complexity adds to the intricacy of understanding the immune response and the potential for developing immunity against different snake venoms.

## 7. Conclusions

In conclusion, the concept of snakebite-induced immunity continues to be a subject of scientific interest and investigation. While there is evidence of protection or altered immune responses in individuals with a history of snakebites, especially in indigenous communities, further research is needed to fully understand the detailed mechanisms and implications of such an immunity. Until then, the primary focus should remain on prevention, prompt medical intervention, and the availability of effective antivenom therapies to mitigate the devastating effects of snakebite envenomation.

## Figures and Tables

**Figure 1 toxins-15-00638-f001:**
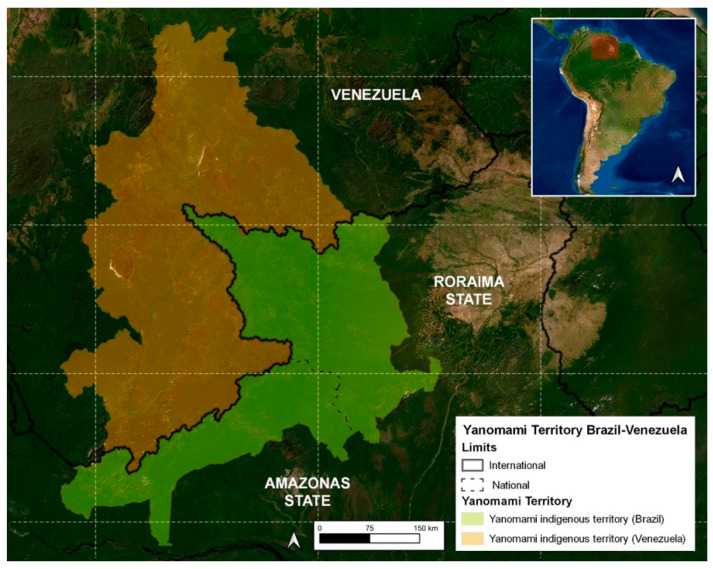
Map of the Yanomami indigenous land.

**Figure 2 toxins-15-00638-f002:**
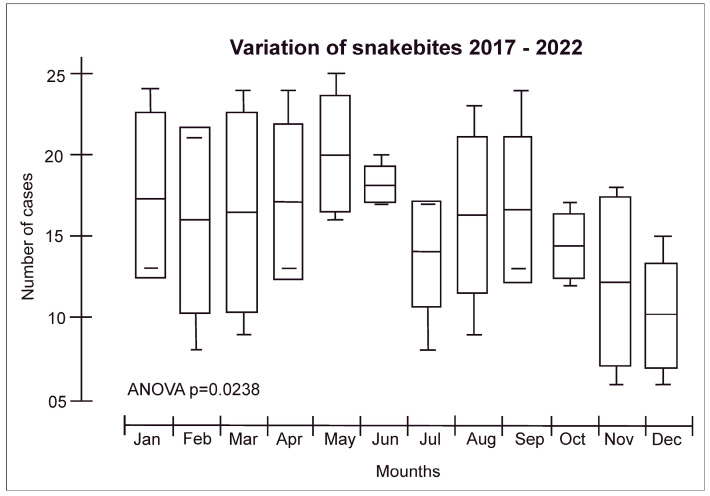
Snakebite distribution according to seasonality in the Yanomami Indigenous Land. The seasonality in Roraima is represented by a rainy period (from April to September) and a dry period (from October to March).

**Figure 3 toxins-15-00638-f003:**
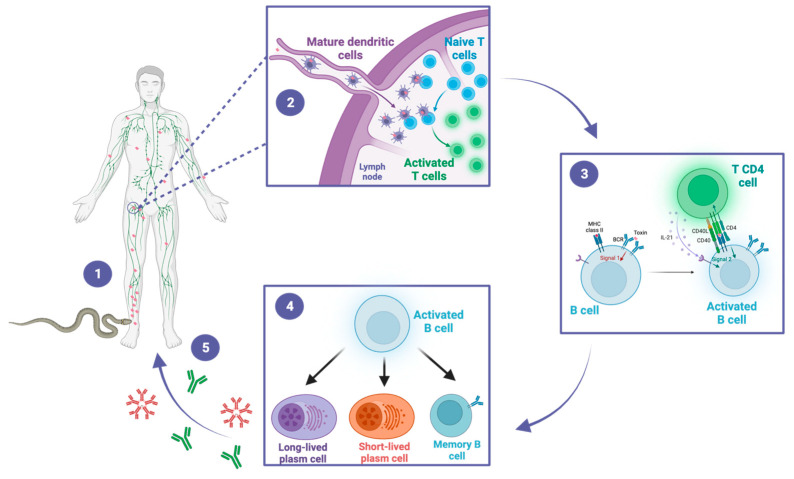
A snakebite inducing antivenom antibodies. This figure illustrates the sequential steps of the immune response to snakebite, highlighting the crucial roles of dendritic cells, T cells, and B cells in the generation of protective antibodies. (1) The snake bites the victim and inoculates the venom, composed of a cocktail of toxins. The toxins are distributed locally and systemically throughout the body. Dendritic cells recognize and process the venom-derived toxins and migrate to the draining lymph nodes. (2) In the lymph nodes, dendritic cells present toxin-derived peptides to naïve CD4+ T cells, leading to their activation. (3) Activated CD4+ T cells in turn activate B cells through a T-dependent pathway. (4) Activated B cells differentiate into memory B cells, as well as into short-lived or long-lived plasma cells. (5) Short-lived plasma cells generate early IgM and IgG antibodies, while long-lived plasma cells sustain the production of IgG antibodies for long-term immunity. These antibodies could contribute to the protective immunity of snakebite victims during subsequent exposures.

**Figure 4 toxins-15-00638-f004:**
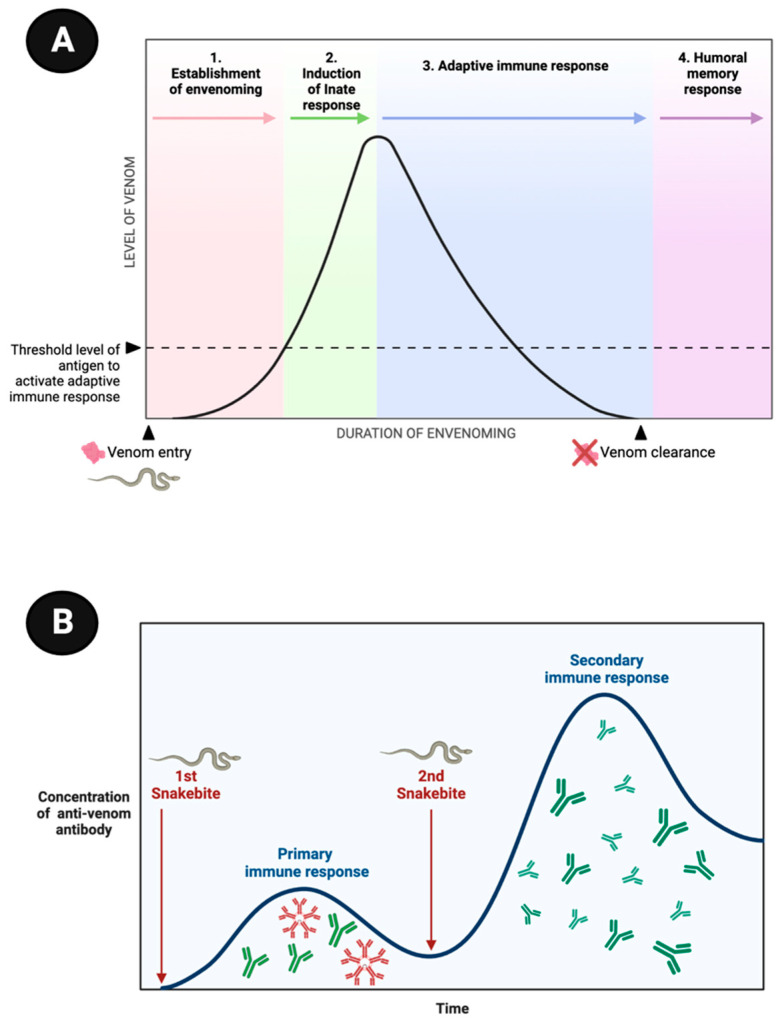
Stages of the humoral memory response following a snakebite. (**A**) Venom-derived toxins (antigens) are rapidly distributed throughout the victim’s body (1) and trigger an immediate innate immune response following the snakebite (2). Conversely, the adaptive immune response takes several days to activate, leading to the production of a humoral response, including specific antibodies, typically occurring 5 to 7 days after the bite (4). (**B**) Initially, the concentration of specific antibodies targeting the venom is low (first episode), consisting primarily of IgM and IgG antibodies. However, upon a subsequent exposure (second episode), the antibody titers increase significantly, with predominantly high levels of IgGs. This enhanced immune response during re-exposure is indicative of a memory response, demonstrating the ability of the immune system to perform a stronger and more rapid defense against the venom. The figures are based on well-established immunological mechanisms [37].

**Table 1 toxins-15-00638-t001:** Studies demonstrating specific antibody production against snake venoms in snakebite victims.

Snake Species	Patients (n)	Country	Main Results	Method of Detection	Antibody Type	Ref.
*Echis carinatus* and *Echis schistosa*	1	United Kingdom	In a patient bitten by two different species (*Echis carinatus* and *E. schistosa*) in different periods (2 years old and 22 years old, respectively), high antivenom antibodies against *E. carinatus* were identified, while a lack of antibodies were identified against *E. schistosa.*	ELISA	-	[42]
Several snakes	223	Ecuador	79% of the patients exhibited antivenom antibodies; most patients had antibodies against more than one snake species.	ELISA	-	[16]
*Echis carinatus*	12	Nigeria	In 10 victims exposed to *Echis carinatus*, it antivenom antibodies were detectable in 4 patients up to 14 days after the bite. Two children previously bitten by *Echis carinatus* were exposed again to the same species and presented a rapid onset of antibodies (<48 h).	ELISA	-	[47]
*Bitis arietans*	1	United Kingdom	After the bite, the patient was followed for 81 days, with weekly sample collection. Significant antibody levels were detected on the 9th day, rising to a peak three weeks after the bite and sustained for 11 days before starting to decrease. The last sample collected on the 81st day displayed a 50% decrease in the antibody level when compared with the antibody titer peak.	ELISA	-	[48]
Several snakes	43	French Guiana	In 43 patients tested for a specific venom antibody, 51% were positive. It was observed that the degree of positivity of the ELISA and the intensity of the immune response was influenced by the degree of symptoms and by the time of exposure. No antibody was detected at periods greater than 15 years.	ELISA	-	[49]
*Hemachatus haemachatus*	1	South Africa	A farm worker with several previously dermatological exposures to *H. huemachatus* demonstrated significant amounts of IgE antibodies against a 66 kDa fraction of the snake venom and high titers of IgG antibody against the same snake, not identified in the control.	ELISA/Western blotting	IgE/IgG	[50]
*Bothrops jararaca*	22	Brazil	Early appearance and short duration of antivenom IgM, observed on the 3rd day post-exposure and disappearing by the 20th day; IgG antibodies became detectable starting from the 18th day and showed a progressive increase until the 80th day; in patients with previous exposure to snake venom, IgG antibodies were detected as early as the 3rd day, with significantly higher levels.	ELISA/Western blotting	IgM/IgG	[45]
*Gloydius blomhoffii*	20 (19 with their first exposure and 1 previously exposed)	Japan	IgG was detected starting from the first week, and the presence of IgG1 and IgG4 was observed even after 15 years of exposure; in patients with prior exposure, a prompt IgG production was observed following the second contact.	ELISA	IgG subclasses (1–4)	[51]
*Naja kaouthia*	50	Thailand	38 patients exhibited antibodies against the venom; a distinct group of 8 patients demonstrated the presence of specific antibodies targeting the α-neurotoxin.	EIA	-	[52]
*Daboia russelli siamensis*	158	Myanmar	95.5% of patients exhibited antivenom antibodies; IgM specific to the venom was detectable as early as the 3rd day following exposure, reaching its peak on the 7th day, and becoming undetectable after 6 weeks (42 days). Patients who developed uremia showed a comparatively lower production of IgM; IgG was detected from the first week onward, indicating a relatively rapid and sustained immune response.	EIA	IgM/IgG	[46]
*Ophiophagus hannah* (King Cobra)	2	Myanmar	Venom IgM antibody was produced a week earlier than venom IgG antibody, peaked at day 8 and fell to a low base within 12–16 days after the bite. However, the venom IgG antibody presented a different development in each patient: the first one, with no previous contact with the species and treated with antivenom, developed a short peak in 12th day followed by a quick fall between the 16–28th day. The second one had received small doses of venoms, including from a king cobra, for years as a method of traditional immunization and did not receive antivenom after the exposure. In this subject, the venom IgG had a sustained response, reaching its highest concentration in oday 12 and maintaining a plateau for two weeks.	EIA	IgM/IgG	[53]
*Crotalus durissus terrificus*	16	Brazil	IgM and IgG were evident by the 3rd day, with IgM reaching its peak titer production between the 7th and 18th day, and IgG levels peaking between 30–90 days	ELISA	IgM/IgG	[44]
*Bothrops alternatus* and *Bothrops pubescens*	11	Uruguay	Comparing the immune response against both snakes, a higher production of IgM antibodies against *B. pubescens* venom was noted compared to those from *B. alternatus* venom. Similar concentrations of IgG were observed among the two groups.	ELISA/Western blotting	IgM/IgG	[54]

ELISA: enzyme-linked immunosorbent assay; EIA: enzyme immunoassay.

## Data Availability

Not applicable.

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
