# Peer review of "The State-of-the-Art of the Humoral Memory Response to Snakebites: Insights from the Yanomami Population"

_toxins, 2023, doi:10.3390/toxins15110638_

Round 1
Reviewer 1 Report
Comments and Suggestions for Authors
The work entitled “The State-of-Art of Humoral Memory Response to Snakebites: 2 Insights from the Yanomami Population” reviews a significant part of the literature regarding snakebite envenomation – induced immunity in humans. In general, the authors provide a good overview of the subject and discuss some of the more relevant matters. Nonetheless some important things, such as differentiation between animal and human studies, and differences in venom inoculation from the snakes are not sufficiently discussed.
Mayor considerations:
Line 83. The final statement of this paragraph uses the word “evolution” in a very ambiguous way. Since the authors don’t cite or discuss any evolutionary process at play in human populations, I suggest deleting this line or discussing in a clear way what is meant by “human evolution”.
Line 86. It is not clear if this study is published elsewhere or if the authors conducted this research. Please clarify.
Line 159. The description of the immunological response provided is very general, but it is treated as if it was a snakebite-specific process. Please clarify that this process occurs with any external protein or pathogen and specify if there are any processes that are specific to snakebite cases.
Figure 4. The figures are too general, and it is not clear if they are based on any data. Please clarify or cite any data supporting the figures.
Table 1. I suggest adding a column with the detection method used and what antibodies were measured (Class and target).
Line 267. Why is this assumed to be caused by cutaneous exposure instead of inhalation of dry venom? Is there any evidence that dermatological exposure was the main factor related to the individual’s immune response? Please clarify.
Line 348. The authors cite a big study, but they do not mention if there were any conclusions from it. Please state if there are any main observations derived from this study.
Finally, there is no discussion regarding possible variation in venom yields from one snake to the other or the possibility of venom not being injected during a bite (dry bites). Please add a small discussion on this subject since it could be central to the conclusions of many of the cited studies.
Minor revisions.
Line 45. Replace “human specific antibodies” for “human antibodies” since the first one is unclear and could mean antibodies against human proteins or targets.
Line 39. I suggest deleting the word “permanent” since the possibility of permanent protection against snakebite is seldom discussed. The authors only review the possibility of this protection existing.
Line 112. Add citation or clarify where the figure is from.
Line 151. Since this paragraph is describing the Yanomami way of life, I suggest moving it before the description of the snakebite accidents.
Line 183. Correct citation
Line 195. The last line is stated as a fact and that can be very misleading. I suggest changing to “These antibodies could contribute to the protective immunity of snakebite victims…”
Line 264. The first sentence is incomplete.
Comments on the Quality of English LanguageLine 63. Consider replacing “in loco” for “on-site” or “in situ” for clarity.
Line 64. Replace “working in area” with “working in the area”.
Line 80. Replace the word “believes” with “beliefs”.
Line 103. Replace “shown” with “showed”.
Line 187. Replace “bite” with “bites”.
Line 265. Replace “tites” with “titles”
Author Response
RESPONSE TO REVIEWERS
Manuscript ID: toxins-2680648
Toxins
Editors-in-Chief: Prof. Dr. Jay Fox
Title: The State-of-Art of Humoral Memory Response to Snakebites: Insights from the Yanomami Population
Thank you very much for your considerable effort in reviewing our manuscript. It is also appreciated that you considered our work of interest for your journal and its readers to allow the submission of a revised version. It stimulated us to amend the text to meet your constructive comments. In what follows, you will find a point-by-point list of how we dealt with reviewer comments in blue, and necessary changes are highlighted in the manuscript in red. We hope that this version is now acceptable for publication in Toxins.
Comments of the Reviewers:
Reviewer 1:
The work entitled “The State-of-Art of Humoral Memory Response to Snakebites: Insights from the Yanomami Population” reviews a significant part of the literature regarding snakebite envenomation – induced immunity in humans. In general, the authors provide a good overview of the subject and discuss some of the more relevant matters. Nonetheless some important things, such as differentiation between animal and human studies, and differences in venom inoculation from the snakes are not sufficiently discussed.
Mayor considerations:
Line 83. The final statement of this paragraph uses the word “evolution” in a very ambiguous way. Since the authors don’t cite or discuss any evolutionary process at play in human populations, I suggest deleting this line or discussing in a clear way what is meant by “human evolution”.
Response: This statement was deleted.
Line 86. It is not clear if this study is published elsewhere or if the authors conducted this research. Please clarify.
Response: This study was performed by Prof. Pucca’s research group, which now is detailed in the manuscript.
“A detailed clinical-epidemiological data obtained from the Yanomami and Ye’kuana Indigenous Special Health District (DSEI-YY) from 2017 to 2022 conducted in accordance with the Declaration of Helsinki (protocol approved by the Research Ethic Committee of the Federal University of Roraima under number CAAE 53970721.4.0000.5016), per-formed by the research group and presented for the first time here, ….”
Line 159. The description of the immunological response provided is very general, but it is treated as if it was a snakebite-specific process. Please clarify that this process occurs with any external protein or pathogen and specify if there are any processes that are specific to snakebite cases.
Response: We appreciate your observation. Our intention was to elucidate the immunological mechanisms responsible for inducing immunity to a non-specialized audience. We have now incorporated this crucial information into the manuscript.
“After a snakebite and the subsequent introduction of venom, the immune system is initiated in accordance with a well-established pattern, leading to the activation of both innate and adaptive immune responses. It's worth emphasizing that this immune mechanism is applicable to any other external or extracellular antigen”
Figure 4. The figures are too general, and it is not clear if they are based on any data. Please clarify or cite any data supporting the figures.
Response: The figures in our manuscript depict the stages of the humoral memory response following snakebite, providing an overview of the immune processes involved. The data supporting these figures are derived from established research in the field, although we understand the need for clarity and citation. The info required was added in the legend: “The figures are based on well-established immunological mechanisms [37]”
Table 1. I suggest adding a column with the detection method used and what antibodies were measured (Class and target).
Response: These columns were added to the table.
Line 267. Why is this assumed to be caused by cutaneous exposure instead of inhalation of dry venom? Is there any evidence that dermatological exposure was the main factor related to the individual’s immune response? Please clarify.
Response: We apologize for the misinterpretation. Indeed, the article discusses exposure, but it is not possible to ascertain whether the contact was topical or inhalatory; it simply rules out snakebite. The text has been amended in the manuscript.
Line 348. The authors cite a big study, but they do not mention if there were any conclusions from it. Please state if there are any main observations derived from this study.
Response: The study demonstrated that the generated immune response was able to prevent local damage caused by the venom. A better and detailed explanation was added in the manuscript.
“Meticulous data collection and analysis revealed that the immunization successfully prevented the occurrence of local lesions in vaccinated individuals exposed to the venom, highlighting the success of large-scale efforts in addressing the devastating impacts of snakebite envenomation and provide valuable insights for future research and the de-velopment of effective immunization strategies”.
Finally, there is no discussion regarding possible variation in venom yields from one snake to the other or the possibility of venom not being injected during a bite (dry bites). Please add a small discussion on this subject since it could be central to the conclusions of many of the cited studies.
Response: These suggestions were added in the manuscript (lines 378-385).
Minor revisions.
Line 45. Replace “human specific antibodies” for “human antibodies” since the first one is unclear and could mean antibodies against human proteins or targets.
Line 39. I suggest deleting the word “permanent” since the possibility of permanent protection against snakebite is seldom discussed. The authors only review the possibility of this protection existing.
Response: These revisions were added in the manuscript.
Line 112. Add citation or clarify where the figure is from.
Response: Figure 2 is a result from a study of the group, performed at Federal University of Roraima (under protocol number CAAE 53970721.4.0000.5016), which was detailed in the manuscript.
Line 151. Since this paragraph is describing the Yanomami way of life, I suggest moving it before the description of the snakebite accidents
Line 183. Correct citation
Line 195. The last line is stated as a fact and that can be very misleading. I suggest changing to “These antibodies could contribute to the protective immunity of snakebite victims…”
Line 264. The first sentence is incomplete.
Response: These revisions were added in the manuscript.
Comments on the Quality of English Language
Line 63. Consider replacing “in loco” for “on-site” or “in situ” for clarity.
Line 64. Replace “working in area” with “working in the area”.
Line 80. Replace the word “believes” with “beliefs”.
Line 103. Replace “shown” with “showed”.
Line 187. Replace “bite” with “bites”.
Line 265. Replace “tites” with “titles”.
Response: All these inputs were added in the manuscript.
Reviewer 2 Report
Comments and Suggestions for Authors
The authors confirmed the concept and profiles of snakebite-induced immunity based on some scientific evidence, and also suggest to conduct further research for fully understand the detailed mechanisms and implications of such immunity. This review will help the readers to well understand the humoral memory response to snakebites.
I suggest the authors to conduct some minor improvment.
1, line265: "tites" change to "titers".
2, Although the vaccination strategy were described in some previous studies, it would not be suitable for treatment of snakebite risks. Actually, the incidence of snakebite is relatively lower than other epidemic disease, and specific vaccination strategies have to be desinged for the snakebites caused by different venomous snakes, and thus result in high cost-effectiveness. I suggest the authors to state the detailed limitation of the vaccination strategy in treatment of snakebite risks.
3, line364: I think snakebite immunity is a fact. And I suggest the subtitle "The snakebite immunity: fact or fiction?" change to "The snakebite immunity: a fact lacking precise assessment".
Author Response
RESPONSE TO REVIEWERS
Manuscript ID: toxins-2680648
Toxins
Editors-in-Chief: Prof. Dr. Jay Fox
Title: The State-of-Art of Humoral Memory Response to Snakebites: Insights from the Yanomami Population
Thank you very much for your considerable effort in reviewing our manuscript. It is also appreciated that you considered our work of interest for your journal and its readers to allow the submission of a revised version. It stimulated us to amend the text to meet your constructive comments. In what follows, you will find a point-by-point list of how we dealt with reviewer comments in blue, and necessary changes are highlighted in the manuscript in red. We hope that this version is now acceptable for publication in Toxins.
Comments of the Reviewers:
Reviewer 2:
The authors confirmed the concept and profiles of snakebite-induced immunity based on some scientific evidence, and also suggest to conduct further research for fully understand the detailed mechanisms and implications of such immunity. This review will help the readers to well understand the humoral memory response to snakebites.
I suggest the authors to conduct some minor improvment.
1, line265: "tites" change to "titers".
Response: This changed was added in the manuscript.
2, Although the vaccination strategy were described in some previous studies, it would not be suitable for treatment of snakebite risks. Actually, the incidence of snakebite is relatively lower than other epidemic disease, and specific vaccination strategies have to be desinged for the snakebites caused by different venomous snakes, and thus result in high cost-effectiveness. I suggest the authors to state the detailed limitation of the vaccination strategy in treatment of snakebite risks.
Response: Limitations regarding vaccination were added in the manuscript (lines 362-371).
3, line364: I think snakebite immunity is a fact. And I suggest the subtitle "The snakebite immunity: fact or fiction?" change to "The snakebite immunity: a fact lacking precise assessment".
Response: Section title was modified.